# Comparative Analysis of Re-Annotated Genes Provides Insight into Evolutionary Divergence and Expressions of Aquaporin Family in Pepper

**DOI:** 10.3390/plants10061039

**Published:** 2021-05-21

**Authors:** Yeon Mi Lee, Geun Young Chae, Min Kyung Kim, Seungill Kim

**Affiliations:** Department of Environmental Horticulture, University of Seoul, Seoul 02504, Korea; codsuak@gmail.com (Y.M.L.); jennygychae@gmail.com (G.Y.C.); 1004mnin@naver.com (M.K.K.)

**Keywords:** aquaporin, water transport, re-annotation, pepper, abiotic stress

## Abstract

Aquaporins (AQPs) are known to have a vital role in water transport in all living organisms including agriculturally important crops, but a comprehensive genomic study of AQPs in pepper has not been implemented. Here, we updated previous gene annotations and generated a total of 259 AQP genes from five plants, including pepper. Phylogenetic and motif analyses revealed that a large proportion of pepper AQP genes belong to the specific subgroup of tonoplast intrinsic protein (TIP) subfamily, TIP4. Chromosomal localization and estimated duplication times illustrated that genes in TIP4 formed a tandem array on the short arm of chromosome 1, resulting from pepper-specific expansion after its divergence with *Solanaceae* species. Transcriptome analyses under various abiotic stress conditions revealed that transport-, photosystem-, and thylakoid-related genes were generally enriched in expression clusters containing AQP genes in pepper. These results provide valuable genomic resources and insight into the evolutionary mechanism that generate genomic diversity of the AQP gene family in pepper.

## 1. Introduction

Aquaporins (AQPs) are small transmembrane proteins (21–34 KDa) from a large family of major intrinsic proteins (MIPs) that facilitate the movement of water, glycerol, and small molecules such as carbon dioxide (CO_2_) and hydrogen peroxide (H_2_O_2_) through biological membranes [1,2]. In plants, the AQP gene family is responsible for the selective transport of water and regulates the response to abiotic stresses such as drought and saline conditions [3]. For example, AtPIP2;2 transports water in *Arabidopsis* [4], and SlTIP2;2 increases the transpiration of tomato under drought and salt stresses [5]. Aquaporins typically have six transmembrane α-helices (H1 to H6) that are connected by five spiral loops (LA-LE). Loops B and E have two conserved asparagine–proline–alanine (NPA) motifs. Groups of amino acid residues, two groups in helices 2 and 5 (H2 and H5) and two groups in loop E (LE1 and LE2), form the aromatic/Arg (ar/R) selectivity filter. The AQP gene family in plants is classified into five major subfamilies: plasma membrane intrinsic proteins (PIPs), tonoplast intrinsic proteins (TIPs), NOD26-like intrinsic proteins (NIPs), small basic intrinsic proteins (SIPs), and uncategorized X intrinsic proteins (XIPs) [6].

Hot pepper (*Capsicum annuum*) is an economically important crop belonging to the *Solanaceae* family with tomato and potato. It is not only a source of the popular spice capsaicin but is of nutritional value as a source of vitamins, minerals, and phenolic compounds [7,8,9]. According to the United Nations Food and Agricultural Organization (FAO, Rome, Italy), world production of hot pepper increased by about 28% over the last decade, and in 2019, the top 10 pepper-producing countries cultivated 32 million tons (http://www.fao.org/home/en/ (accessed on 2 December 2020)). Recently, pepper genome sequencing projects with transcriptome studies have enabled identification and characterization of important gene families related to agricultural traits [8,10,11,12]; however, comprehensive understanding of AQP genes in pepper through integrative genomic and transcriptome studies has not been achieved.

In this study, we updated the annotation of AQP gene family including AQP-like as well as AQP genes in pepper, tomato, potato, rice, and *Arabidopsis* and performed a comparative analysis. Of the five species we studied, pepper had the most AQP genes, and 42% of these were in the TIP subfamily. Specifically, from phylogenetic analysis, we determined that most of these pepper genes (74%) were in the TIP4 subgroup. We also investigated the AQP domain motif composition of each subfamily of these five species. Motifs 7, 19, and 5, located in downstream positions 11, 12, and 13, respectively, were specific to the TIP4 subgroup of pepper. We demonstrated, with chromosomal localization, that 52% of the pepper AQP genes in the TIP4 subgroup clustered in the short arm of chromosome 1. Most of these genes had been duplicated after the divergence time between pepper and *Solanaceae* species. Expression clustering and gene ontology (GO) enrichment analyses demonstrated that genes with putative functions related to chloroplast, thylakoid, and transport were enriched in clusters containing the pepper AQP genes, which may be evidence that these AQP genes work with other pepper genes in the plant response to abiotic stresses, especially in the transport system of thylakoid membranes.

We anticipate that these updated AQP gene models will provide novel genomic resources to horticulturalists and enrich understanding of recent evolutionary divergence in pepper AQP genes and the complex transcriptional network that operates in abiotic stresses.

## 2. Results

### 2.1. Re-Annotation and Classification of AQP Genes in Pepper and Other Species

To reduce annotation bias from different annotation methods with resources [13], we performed re-annotation and generated a total of 259 AQP genes containing 29 (11.2%) newly annotated genes in addition to previously annotated genes (Table 1). In the updated AQP genes, we observed 73 pepper AQP genes that were 1.8-fold of AQP genes in rice and *Arabidopsis* genomes. In addition, numbers of AQP genes in other *Solanaceae* genomes were relatively higher than those of rice and *Arabidopsis* (Table 1).

We classified AQP genes into five subfamilies by comparing specific amino acid sequences in ar/R selectivity filter positions among five genomes (Table 2 and Appendix A). A large number of the pepper AQP genes belonged to the TIP subfamily, approximately threefold more than in the other species. Specifically, 31 pepper AQP genes (42.5%) were in the TIP subfamily, suggesting that specific AQP genes have been expanded in the pepper genome (Table 1). We also observed many genes belong to the XIP subfamily in potato and tomato genomes, but none in *Arabidopsis*; this is consistent with a previous report [14,15]. These results demonstrated that the copy numbers of AQP genes in the five plant genomes have changed via proliferation of genes belonging to specific subfamilies in the *Solanaceae*, especially in pepper.

We investigated the motif composition of AQP domains to characterize the structural diversity of these domains, which are mostly conserved and essential for transport function of those genes (Figure 1 and Appendix A, Appendix A). Overall, motif compositions were similar in each subfamily among the five plant species. Specifically, the motifs at the N- and C-terminal regions were mostly conserved among subfamilies, whereas the central regions, which includes the subfamily-specific motifs with ar/R selectivity filter position, were relatively variable. Thus, it may be that the motifs in the central account for the structural diversity of AQP domains observed among subfamilies (Figure 1 and Appendix A). When we examined the motifs of AQP domains in pepper, we observed the specific motifs (7, 19, 5) were enriched in the TIP subfamily compared to the other species (Figure 1 and Appendix A). This suggests that those motifs contributed to the structural diversity of pepper AQP domains in the TIP subfamily.

### 2.2. Phylogenetic Analysis of Plant AQP Genes

We next performed a phylogenetic analysis and determined the subgroups (of the subfamilies) to which the AQP genes belonged. The subfamilies PIP, TIP, and NIP were further separated into three, four, and three subgroups, respectively, whereas SIP and XIP remained as single groups (Figure 2A). Notably, a large number of AQP pepper genes in the TIP subfamily were mostly in the TIP4 subgroup—23 genes, representing 74% of the pepper genes in the TIP subfamily (Figure 2B). Examination of the motif composition of AQP domains in TIP subgroups revealed a set of specific motifs, 7, 9, and 15 on downstream regions, were only observed in pepper AQP domains of TIP4 (Figure 2C). These results are consistent with proliferation of specific TIP genes in pepper genome via lineage-specific duplication, which led to expansion of pepper TIP genes (Figure 2A).

### 2.3. Chromosomal Location and Gene Duplication Analyses of Plant AQP Genes

To identify the location of AQP genes, we mapped their physical positions on the chromosomes of each genome (Figure 3 and Appendix A). Overall, AQP genes were distributed across all chromosomes, with a few exceptions. We mapped 64 AQP genes in pepper, which has 12 chromosomes, and 33% of them mapped to chromosome 1 (Figure 3). More specifically, 12 of the pepper AQP genes in TIP4 consisted of a tandem array on the short arm of chromosome 1 (Figure 3). Considering that there were no TIP4 genes in the syntenic regions of the tomato or potato chromosomes, this suggests that these 12 pepper genes emerged recently as a result of pepper-specific gene duplication. We also noted that seven potato and five tomato AQP genes belonging to the XIP1 subgroup formed clusters on the long arm of chromosome 10, which was specific to the *Solanum* spp. (Appendix A).

We next explored the duplication history of AQP genes. Gene duplication time was estimated by calculating the synonymous substitution rates (Ks) between recently duplicated gene pairs (Figure 4). Our analysis showed that the pepper AQP genes, especially those in the TIP4 subgroup, were actively duplicated, with a maximum Ks value of approximately 0.15. This also suggests that most of the duplication of the pepper AQP genes occurred after the divergence between pepper and other *Solanum* species, which have a Ks value of 0.3 [8] (Figure 4B,C). Taken together, these results indicate that recent gene duplications mainly occurred in TIP4 subgroup was a major contributor for construction of diversified AQP gene repertories in the pepper genome through expansion of specific AQP genes.

### 2.4. Expression Profiles and Clustering Analyses of Pepper AQP Genes from Plants Grown under Abiotic Stress

Plants adapt to stressful environments through various systemic physiological changes. We examined responses to abiotic stresses in pepper at the level of transcription and expression. We first established an expression profile for the AQP genes under cold, heat, salt, and mannitol stresses (Appendix A). Overall, a few genes (14) belonging to the PIP subfamily were highly expressed in all stress conditions, whereas several genes (14) belonging to the TIP subfamily were not, with one exception: CaTIP4;10 was highly expressed in all stress treatments, suggesting that it may have a particular function in various abiotic stresses. To group pepper genes with parallel expression patterns and thus predict genes with similar functions, we then performed expression clustering analyses and identified 18,052 (50%), 17,438 (48.6%), 15,926 (44.3%), and 16,105 (45%) DEGs in plants subjected to cold, heat, salt, and mannitol stress, respectively. Among these DEGs, 36, 42, 35, and 35 were AQP genes in cold, heat, salt, and mannitol, respectively (Figure 5A). The clusters containing the most AQP genes were the cold cluster 5 (36%, 13 genes), the heat-cluster 4 (26%, 11 genes), the salt cluster 6 (29%, 10 genes), and the mannitol cluster 1 (26%, 9 genes) (Figure 5B). Most of the differentially expressed AQPs were in the PIP subfamily, and all gene clusters from plants exposed to heat and mannitol treatments contained PIP genes. The plants subjected to heat stress had 15 genes in the TIP subfamily, more than plants exposed to the other stresses.

Furthermore, functional enrichment analysis, performed to predict the putative gene functions, revealed that genes with chloroplast-, photosystem-, organelle-, and thylakoid-related GO terms were generally enriched in the clusters including AQP genes, suggesting that those genes could be involved in response to various abiotic stresses along with AQP genes (Figure 5C). Specifically, many plants use osmotic adjustment and ion transport through AQP [16] to adapt to stress, and the finding that anion and organic substance transport-related GO terms were enriched in the salt cluster 5 (Figure 5C) supports this suggestion. Moreover, enrichment of GO terms related to the thylakoid membrane in cold cluster 5 were consistent with previous studies, which showed that the TIP genes in *Arabidopsis* control the volume of the thylakoid lumen during water or light stress, thereby creating an optimal photosynthetic environment [17]. Our results suggest that the pepper AQP genes also have an important role in molecular responses to changing conditions and act in concert with other pepper genes.

## 3. Discussion

AQP is an important gene family involved in water transport and other small molecules such as CO_2_ and H_2_O_2_ plants, but the complete understanding of the AQP gene family in pepper has not been accomplished. For in-depth studies, an accurate annotation process involving structural annotation to fully predict gene structures containing target domains within the genome sequence is essential to provide a stronger foundation for biological investigation of living organisms [18]. However, reports of missing functional genes [19] or incomplete gene models [13] continue to be registered, generating bias in downstream studies.

In this study, we re-annotated AQP genes in five plant genomes using the target gene family annotation approach [20] to update AQP gene information via detection of previously omitted AQP genes. In general, whole-gene annotation methods, which have been applied to various species including model plants, have pre-processing steps such as repeat-masking to efficiently reduce computing power and manual work [21,22]. However, these steps often generate omission of genes in annotations [13], which are genes that exist in the assembled genome but are obscured by repeat-masking or unidentified due to other problems. Target gene family annotation methods were developed to complement these issues, and we conducted re-annotation using TGFam-Finder, developed to identify all target genes in assembled genome via intensive annotation of genomic regions containing target domain(s) [20].

Prior studies have also reported improvement of annotations of model plants such as *Arabidopsis* and rice, as well as human genome annotation by identification of missing genes in previous annotations [13,20]. Of the updated genes, we newly identified AQP genes omitted in previous annotations including rice and *Arabidopsis* and especially in the pepper genome. The updated AQP genes could serve as genomic resources for further comparative and evolutionary analyses on the basis of their accurate copy numbers in the assembled genomes.

Comparison of the five updated AQP gene models revealed that pepper had the highest number of AQP genes. Many previous studies reported that different copy numbers of gene families between species are mainly derived from species-specific evolution and resulted in genetic diversification [23,24]. We also observed that most pepper AQP genes were contained in the TIP4 subgroup with pepper-specific motif sequences, suggesting that these genes were lineage-specifically expanded in pepper, thereby increasing the genetic diversity in the AQP repertories.

Chromosomal localization and gene duplication history analyses showed that these pepper-specific genes in the TIP4 subgroup formed a tandem array in the short arm of chromosome 1, and the genes mainly emerged after the speciation time between pepper and other *Solanum* spp. A previous study also reported a similar case in which certain genes in the pepper nucleotide-binding, leucine-rich repeat (NLRs) gene family formed a tandem array in a specific chromosome via massive duplication, indicating that it occurred specifically in pepper [25]. Although most of their roles are unknown, a previous study reported that one gene, *Pvr4*, emerged eight million years ago after speciation between pepper and tomato, gaining a specific function in disease resistance for various potyviruses and showing that a pepper-specific NLR gene had an important role [25]. Similarly, the AQP genes in TIP4, which have undergone species-specific duplication after the pepper speciation time, may also possess important characteristics of agricultural value.

We further inspected the forward genetic screen using the expression level of pepper genes to identify those that are potentially involved in responses to different abiotic stress conditions. The genes belonging to the PIP were highly expressed. This was consistent with many reports that PIP is known to function in a variety of crops; for example, in *Acacia auriculiformis*, the AaPIP1-2 gene eliminated ROS and played a positive role in the drop stress; in *Populus*, the overexpression of PIP1;1 induced more sensitivity of transgenic in osmotic conventions; and in rice, the downregulation of PIP2;1 inhibited root growth [26,27,28]. Expression clustering and GO enrichment analyses on the pepper DEGs under abiotic stresses revealed clusters including AQP were generally enriched in GO terms related to the chloroplast, photosystem, organelle, and thylakoid, suggesting that genes involved in these functions may work together with AQP in response to various stresses [29]. Similarly, we observed newly identified genes in various clusters with other pepper genes. For example, CANN_MIP.PGAv.1.6.scaffold855.1 and CANN_MIP.chr06.1 belonging to salt cluster1 and salt cluster3, respectively, could function together with other pepper genes in cellular processes such as cell communication and membrane-bounded organelles under a salt stress condition. Although the actual functions of TIP4 genes in pepper are difficult to elucidate, their potential functions were predicted on the basis of the expression profiling and clustering analyses. Most genes belonging to the TIP4 subgroup were not significantly expressed, which is consistent with other reports that gene expression of newly emerged genes is not actively detected [30]. However, we found one gene, CA.PGAv.1.6.scaffold1432.12, in mannitol cluster2 that had GO enrichment in ribosome-related tasks, suggesting that this gene may also play a role in plant physiological processes related to ribosomes.

The updated AQP genes will be valuable genomic resources for further functional and breeding research in agriculturally valuable crops. Moreover, our findings provide insights into the evolutionary mechanism and expressions of pepper AQP genes that contribute to increasing genetic diversity and unveiling potential functions, respectively.

## 4. Materials and Methods

### 4.1. Re-Annotation of AQP Genes in Five Plant Genomes

To re-annotate putative aquaporin (AQP) genes including AQP-like genes, we first collected genome resources of *Arabidopsis thaliana* [31], *Oryza sativa* [32], *Solanum lycopersicum* [33], *Solanum tuberosum* [34], and *Capsicum annuum* [8], including assembled genomes, annotated genes, and RNA-Seq data (Appendix A). The re-annotation was performed with TGFam-Finder v1.20 considering parameters described previously [20]. We used tsv files generated from the functional annotation of proteins in the five genomes using InterproScan-5 [35] as “TSV_FOR_DOMAIN_IDENTIFICATION”. PF00230 (MIP) from Pfam database was used as the “TARGET_DOMAIN_ID”. Finally, we obtained re-annotated AQP genes as updated annotations for the five plant genomes, including previously annotated genes from downloaded annotations and genes that were newly annotated from the TGFam-Finder.

We assigned gene names for re-annotated AQP genes. For previously annotated genes, if names had been assigned in previous studies, as was the case for the AQP genes in potato and rice, we matched new gene name to the original names. However, genes in *Arabidopsis* and tomato that had been annotated in previous studies that used different annotation versions. To address this problem, we compared the previously annotated AQP genes in updated annotations to their original versions with BLASTP, and the gene names with the highest identity score were used. Pepper has not been previously studied; therefore, all 73 re-annotated genes were given new names. In addition, newly annotated genes of other species were also assigned new names (Table 1, Table 2, and Appendix A).

### 4.2. Classification and Phylogenetic Analysis of AQP Genes

The re-annotated AQP genes of the five species were aligned using MAFFT v7.470 [36], and unnecessary gaps in the alignments were trimmed with trimAL v1.4 (-gappyout) [37]. We obtained information on the ar/R selectivity filter from each of the four positions in H2 (37th position in the aligned sequences), H5 (143rd), LE1 (152nd), and LE2 (158th), on the basis of the location of the conserved NPA motifs in the alignments (Table 2 and Appendix A). For *Arabidopsis*, rice, tomato, and potato, subfamilies were classified according to the predicted amino acids in each of four positions of the protein, as described previously [38,39,40,41]. We classified the subfamilies of pepper AQPs, however, by considering the known ar/R selectivity filter positions of other four species. We compared unclassified genes to the classified AQP genes using BLASTP with the best match score. From this, we identified members of all five subfamilies of AQP genes: PIP, TIP, NIP, SIP, and XIP in the five species (Table 1, Table 2, and Appendix A).

To understand evolutionary relationships, a phylogenetic tree of aligned AQP sequences was constructed by IQ-TREE v2.0.6 (-alrt 1000-B 1000) [42] and visualized using IToL v3.2.317 (http://itol.embl.de/ (accessed on 18 June 2020)) [43]. On the basis of the phylogenetic tree, we classified 5 subfamilies into 12 subgroups.

### 4.3. Motif Analysis of Subfamilies in Pepper Genome

To identify conserved sequences of AQP domains in five plant species, we used MEME suite v5.1.1 (-mod zoops-nmotifs 50-minw 10-maxw 50-objfun se-markov_order 0) (Appendix A) [44]. Then, the motif position in AQP domains were predicted using MAST v5.1.1 [45]. We determined the general 14 position constructing AQP domains using the identified motif sequences by manual inspections, considering sequence homology. We predicted the number of transmembrane helices and position from AQP domain using TMHMM (Figure 1) [46]. Finally, we counted motif frequencies located in the general position and visualized them using ggplot2 [47] in R package.

### 4.4. Location of AQP Genes on Chromosomes in Five the Plant Genomes

Chromosomal distribution of the re-annotated AQP genes was visualized using MapChart v2.32 [48]. Gene names were marked on the map, and members of the same subgroup, determined by the phylogenetic analysis in this study, were identified by the same color.

### 4.5. Estimation of Duplication Time of AQP Genes

We estimated the divergence times of the AQP genes by first identifying duplicated gene pairs with DupGen_Finder pipeline [49]. Next, the gene pairs were aligned with PRANK (-f = fasta-codon) [50]. We calculated the synonymous substitution rates (Ks) of each gene pair with the KaKs_Calculator v2.0 (-m MYN) [51]. Values of Ks < 2 were visualized with ggplot 2 [47] in an R package.

### 4.6. Expressional Clustering and GO Enrichment Analyses in the Pepper Genome

To examine expression profiles of pepper AQP genes, we collected RNA-Seq data for leaf tissues under various abiotic stress (cold, heat, mannitol, and salt) at different times generated from a previous study [52]. We first performed quality trimming with CLC Assembly Cell (CLC Bio, Aarhus, Denmark) to clear sequences of poor quality. Next, we mapped the RNA-Seq to the genome assembly of pepper using HISAT 2 [53] (-dta-x). The mapped reads were quantified and the “Fragment Per Kilobase of transcript per Million mapped reads” (FPKM) values for the pepper AQP genes were calculated with StringTie [54] (-e-B-G). The overall expression levels of the pepper AQP genes in different conditions were visualized with log2 (FPKM + 1) values using pheatmap v1.0.12 [55] software in an R package.

We performed expression clustering of the differentially expressed genes (DEGs) in the whole annotated pepper genome, including the newly annotated AQP genes. Expression clustering was calculated from log2 transformation of fold changes between the FPKM values measured in plants under stress conditions and unstressed control plants. The DEGs that had a *p* value ≤ 0.05 were clustered using the Mfuzz program [56] in an R package. The number of expression clusters was determined by k-means analysis. The GO terms for genes in each cluster were identified using Omicsbox (https://www.biobam.com/omicsbox/ (accessed on 5 January 2021)), and enriched GO terms were determined by Fisher’s exact test (false discovery rates corrected *p*-value ≤ 0.01).

## Figures and Tables

**Figure 1 plants-10-01039-f001:**
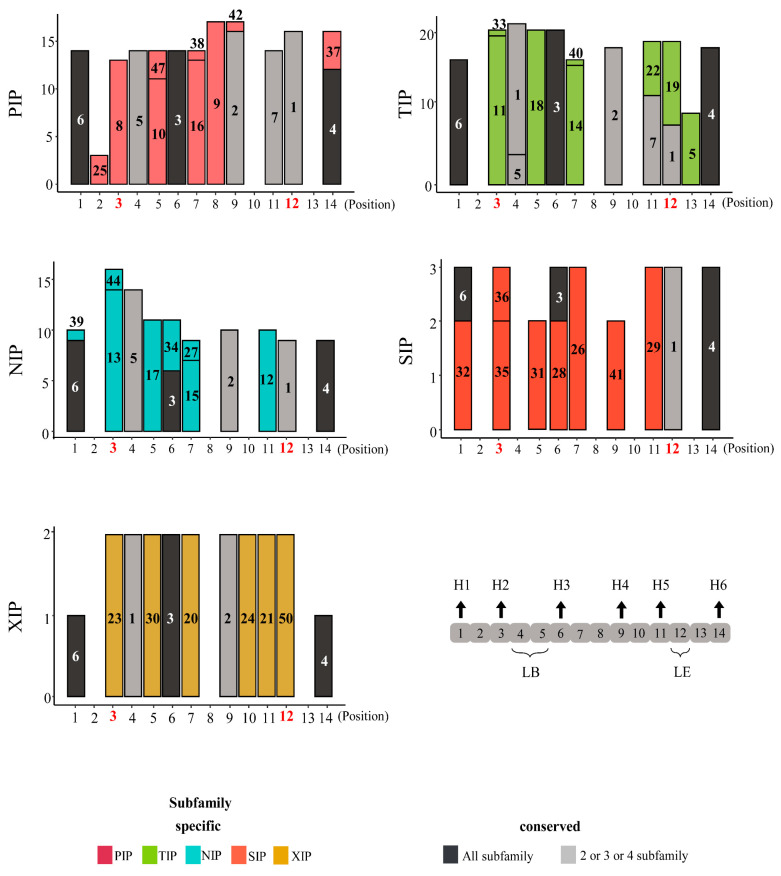
Motif structures for aquaporin (AQP) domains of each subfamily in pepper. The AQP domains consist of a total of 14 positions, and positions highlighted in red represent the location of four residues of ar/R selectivity filter. The *y*-axes indicate the number of genes containing each motif. The motif numbers written in bars indicate motif sequences described in Appendix A. Numbers of motifs in pepper AQP domains are shown as a bar graph. Black arrows show locations of transmembrane helices and loops in motif positions of AQP domains. H1–H6, helix 1 to helix 6; LB, loop B; LE, loop E; PIP, plasma membrane intrinsic protein subfamily; TIP, tonoplast intrinsic protein subfamily; NIP, NOD26-like intrinsic protein subfamily; SIP, small basic intrinsic protein subfamily; XIP, uncategorized X intrinsic protein subfamily.

**Figure 2 plants-10-01039-f002:**
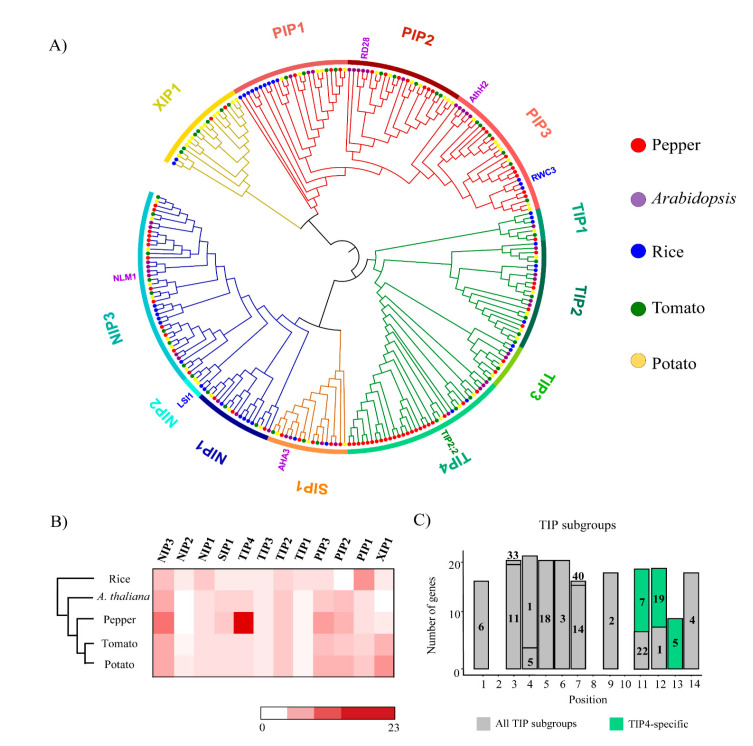
Phylogenetic relationships of aquaporin (AQP) genes in five plant species and conserved motifs of TIP (tonoplast intrinsic protein) in pepper. (**A**) A phylogenetic tree of AQP genes in five plants including pepper is shown. The colored bars on the outer ring represent different subgroups and colored dots at branch tips indicate genes from five species. Gene names written in different colors are known functional genes. (**B**) Numbers of AQP genes from five plant species in each subgroup are shown as a heatmap. (**C**) Numbers of motifs in pepper AQP domains of TIP subgroups are shown as a bar graph. Gray bars are motifs conserved in all clusters and light green bars are subgroup TIP4-specific motifs in the pepper AQP domains.

**Figure 3 plants-10-01039-f003:**
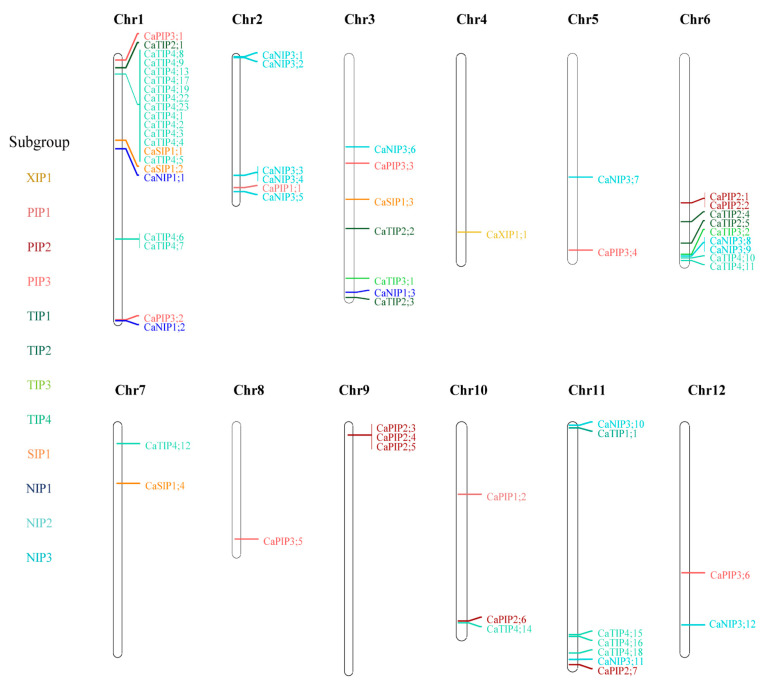
Location of AQP genes on chromosomes in the pepper genome. Colored gene names indicate the different subgroups.

**Figure 4 plants-10-01039-f004:**
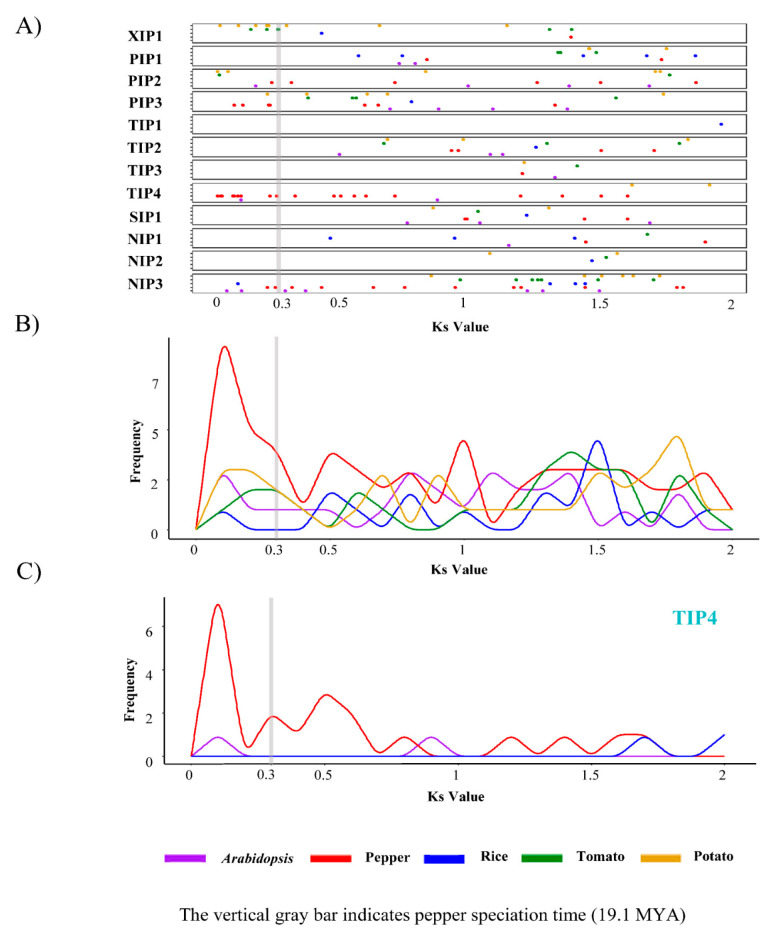
Synonymous substitution rates (Ks) values of duplicated AQP genes in five plant species. (**A**) Ks values between duplication AQP gene pairs in each subgroup are shown as a dot plot. (**B**,**C**) Frequency distribution of Ks values of AQP genes in all subgroups (**B**) and in TIP4 subgroup (**C**). PIP, plasma membrane intrinsic protein subfamily; TIP, tonoplast intrinsic protein subfamily; NIP, NOD26-like intrinsic protein subfamily; SIP, small basic intrinsic protein subfamily; XIP, uncategorized X intrinsic protein subfamily.

**Figure 5 plants-10-01039-f005:**
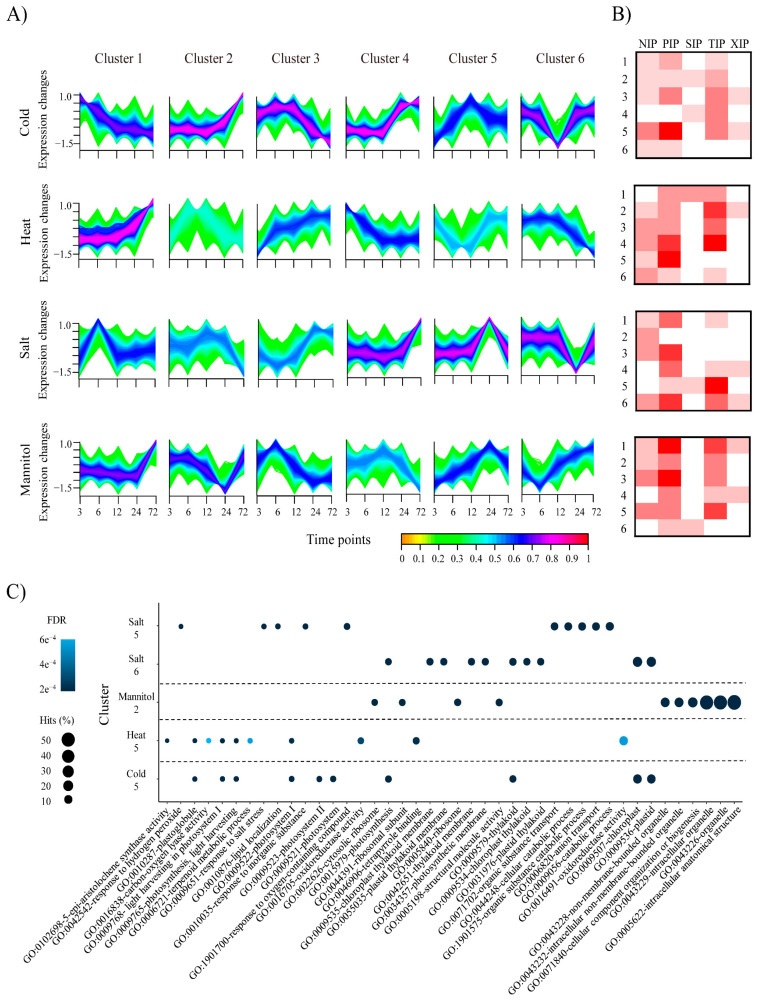
Expression clustering and gene ontology (GO) enrichment analyses of differentially expressed genes (DEGs) in pepper. (**A**) Expression clustering analyses of pepper DEGs from plants exposed to different abiotic stresses are shown. The colored bar shows the membership grade with red indicating high association to each cluster and orange color indicates low association to each cluster. (**B**) Heatmaps showing numbers of pepper DEGs belonging to each AQP subfamily in plants exposed to different abiotic stresses. The heatmap values range from 0 to 6 genes. *Y*-axes present the cluster number in each abiotic condition. (**C**) Top 10 GO enrichment results of representative clusters containing AQP genes are shown as dot plots. Hits show the ratio of the number of genes with the corresponding GO-term to the total number of pepper genes. FDR, false discovery rate; PIP, plasma membrane intrinsic protein subfamily; TIP, tonoplast intrinsic protein subfamily; NIP, NOD26-like intrinsic protein subfamily; SIP, small basic intrinsic protein subfamily; XIP, uncategorized X intrinsic protein subfamily.

**Table 1 plants-10-01039-t001:** Numbers of re-annotated AQP genes in five species.

Common Name	Total (Newly Annotated Genes)	PIPs	TIPs	NIPs	SIPs	XIPs
Rice	41 (3)	13 (-)	11 (-)	13 (1)	2 (-)	2 (2)
*Arabidopsis*	41 (3)	14 (1)	12 (1)	11 (1)	4 (-)	- (-)
Pepper	73 (10)	19 (2)	31 (3)	16 (3)	5 (2)	2 (-)
Potato	56 (9)	19 (3)	11 (-)	13 (1)	3 (-)	10 (5)
Tomato	48 (4)	15 (2)	11 (-)	12 (-)	3 (-)	7 (2)

PIP, plasma membrane intrinsic protein subfamily; TIP, tonoplast intrinsic protein subfamily; NIP, NOD26-like intrinsic protein subfamily; SIP, small basic intrinsic protein subfamily; XIP, uncategorized X intrinsic protein subfamily. The numeric values in parentheses indicate the number of newly annotated genes.

**Table 2 plants-10-01039-t002:** Characterization and classification of individual pepper aquaporin (AQP) genes.

No.	Gene Name	Locus ID	Subfamily	Subgroup	Position (Amino Acid Residues)				Chromosome/ Scaffold Location
					H2 (37)	H5 (143)	LE1 (152)	LE2 (158)	
1	CaPIP1;1	CA.PGAv.1.6.scaffold890.5	PIP	PIP1	F	H	T	R	chr02
2	CaPIP1;2	CA.PGAv.1.6.scaffold216.2	PIP	PIP1	F	H	T	R	chr10
3	CaPIP1;3	CA.PGAv.1.6.scaffold588.86	PIP	PIP1	F	H	T	R	PGAv.1.6.scaffold588
4	CaPIP2;1	CANN_MIP.chr06.1	PIP	PIP2	-	H	T	R	chr06
5	CaPIP2;2	CA.PGAv.1.6.scaffold920.3	PIP	PIP2	F	-	-	-	chr06
6	CaPIP2;3	CA.PGAv.1.6.scaffold888.8	PIP	PIP2	F	H	T	R	chr09
7	CaPIP2;4	CA.PGAv.1.6.scaffold888.7	PIP	PIP2	F	H	T	R	chr09
8	CaPIP2;5	CA.PGAv.1.6.scaffold888.6	PIP	PIP2	F	H	T	R	chr09
9	CaPIP2;6	CA.PGAv.1.6.scaffold1169.2	PIP	PIP2	F	H	T	R	chr10
10	CaPIP2;7	CA.PGAv.1.6.scaffold631.27	PIP	PIP2	F	H	T	R	chr11
11	CaPIP3;1	CA.PGAv.1.6.scaffold774.55	PIP	PIP3	F	H	T	R	chr01
12	CaPIP3;2	CA.PGAv.1.6.scaffold792.78	PIP	PIP3	F	H	T	R	chr01
13	CaPIP3;3	CA.PGAv.1.6.scaffold613.1	PIP	PIP3	L	Y	T	R	chr03
14	CaPIP3;4	CA.PGAv.1.6.scaffold1184.2	PIP	PIP3	F	H	T	R	chr05
15	CaPIP3;5	CA.PGAv.1.6.scaffold78.85	PIP	PIP3	F	H	T	R	chr08
16	CaPIP3;6	CA.PGAv.1.6.scaffold815.3	PIP	PIP3	F	H	T	R	chr12
17	CaPIP3;7	CA.PGAv.1.6.scaffold855.26	PIP	PIP3	F	H	T	R	PGAv.1.6.scaffold855
18	CaPIP3;8	CA.PGAv.1.6.scaffold855.28	PIP	PIP3	-	P	I	G	PGAv.1.6.scaffold855
19	CaPIP3;9	CANN_MIP.PGAv.1.6.scaffold855.1	PIP	PIP3	-	P	I	R	PGAv.1.6.scaffold855
20	CaTIP1;1	CA.PGAv.1.6.scaffold610.69	TIP	TIP1	H	I	A	R	chr11
21	CaTIP2;1	CA.PGAv.1.6.scaffold1058.46	TIP	TIP2	-	I	G	R	chr01
22	CaTIP2;2	CA.PGAv.1.6.scaffold48.31	TIP	TIP2	N	V	G	Y	chr03
23	CaTIP2;3	CA.PGAv.1.6.scaffold438.40	TIP	TIP2	H	I	G	R	chr03
24	CaTIP2;4	CA.PGAv.1.6.scaffold62.35	TIP	TIP2	H	I	G	R	chr06
25	CaTIP2;5	CA.PGAv.1.6.scaffold1152.13	TIP	TIP2	H	I	G	R	chr06
26	CaTIP3;1	CA.PGAv.1.6.scaffold861.16	TIP	TIP3	H	V	A	R	chr03
27	CaTIP3;2	CA.PGAv.1.6.scaffold65.60	TIP	TIP3	H	V	A	R	chr06
28	CaTIP4;1	CA.PGAv.1.6.scaffold298.3	TIP	TIP4	H	T	A	M	chr01
29	CaTIP4;2	CA.PGAv.1.6.scaffold298.4	TIP	TIP4	H	T	A	M	chr01
30	CaTIP4;3	CA.PGAv.1.6.scaffold298.5	TIP	TIP4	-	T	A	M	chr01
31	CaTIP4;4	CANN_MIP.chr01.3	TIP	TIP4	H	-	Q	F	chr01
32	CaTIP4;5	CA.PGAv.1.6.scaffold298.6	TIP	TIP4	M	V	Y	G	chr01
33	CaTIP4;6	CA.PGAv.1.6.scaffold777.2	TIP	TIP4	H	-	-	-	chr01
34	CaTIP4;7	CA.PGAv.1.6.scaffold777.1	TIP	TIP4	H	L	A	M	chr01
35	CaTIP4;8	CANN_MIP.chr01.1	TIP	TIP4	H	T	A	M	chr01
36	CaTIP4;9	CA.PGAv.1.6.scaffold1381.7	TIP	TIP4	H	N	A	M	chr01
37	CaTIP4;10	CA.PGAv.1.6.scaffold1432.12	TIP	TIP4	H	I	A	V	chr06
38	CaTIP4;11	CA.PGAv.1.6.scaffold394.19	TIP	TIP4	H	I	A	V	chr06
39	CaTIP4;12	CA.PGAv.1.6.scaffold467.1	TIP	TIP4	H	I	A	G	chr07
40	CaTIP4;13	CANN_MIP.chr01.2	TIP	TIP4	H	T	A	M	chr01
41	CaTIP4;14	CA.PGAv.1.6.scaffold488.29	TIP	TIP4	H	I	A	V	chr10
42	CaTIP4;15	CA.PGAv.1.6.scaffold820.9	TIP	TIP4	-	I	A	M	chr11
43	CaTIP4;16	CA.PGAv.1.6.scaffold420.10	TIP	TIP4	H	V	A	L	chr11
44	CaTIP4;17	CA.PGAv.1.6.scaffold1381.8	TIP	TIP4	H	T	A	M	chr01
45	CaTIP4;18	CA.PGAv.1.6.scaffold280.18	TIP	TIP4	H	I	A	M	chr11
46	CaTIP4;19	CA.PGAv.1.6.scaffold1381.9	TIP	TIP4	H	T	A	M	chr01
47	CaTIP4;20	CA.PGAv.1.6.scaffold963.13	TIP	TIP4	-	I	A	M	PGAv.1.6.scaffold963
48	CaTIP4;21	CA.PGAv.1.6.scaffold1331.1	TIP	TIP4	H	-	-	G	PGAv.1.6.scaffold1331
49	CaTIP4;22	CA.PGAv.1.6.scaffold298.1	TIP	TIP4	H	T	A	M	chr01
50	CaTIP4;23	CA.PGAv.1.6.scaffold298.2	TIP	TIP4	H	T	A	M	chr01
51	CaNIP1;1	CA.PGAv.1.6.scaffold38.2	NIP	NIP1	A	V	-	-	chr01
52	CaNIP1;2	CA.PGAv.1.6.scaffold792.10	NIP	NIP1	A	I	G	R	chr01
53	CaNIP1;3	CA.PGAv.1.6.scaffold407.53	NIP	NIP1	S	I	A	R	chr03
54	CaNIP3;1	CA.PGAv.1.6.scaffold1545.1	NIP	NIP3	W	V	A	R	chr02
55	CaNIP3;2	CA.PGAv.1.6.scaffold1067.9	NIP	NIP3	W	-	-	-	chr02
56	CaNIP3;3	CA.PGAv.1.6.scaffold411.30	NIP	NIP3	W	V	A	R	chr02
57	CaNIP3;4	CA.PGAv.1.6.scaffold411.29	NIP	NIP3	W	S	A	R	chr02
58	CaNIP3;5	CA.PGAv.1.6.scaffold625.87	NIP	NIP3	W	V	A	R	chr02
59	CaNIP3;6	CANN_MIP.chr03.1	NIP	NIP3	W	-	-	-	chr03
60	CaNIP3;7	CANN_MIP.chr05.1	NIP	NIP3	W	-	-	R	chr05
61	CaNIP3;8	CA.PGAv.1.6.scaffold65.202	NIP	NIP3	W	-	-	K	chr06
62	CaNIP3;9	CA.PGAv.1.6.scaffold65.203	NIP	NIP3	W	I	A	R	chr06
63	CaNIP3;10	CA.PGAv.1.6.scaffold1298.6	NIP	NIP3	-	V	A	R	chr11
64	CaNIP3;11	CANN_MIP.chr11.1	NIP	NIP3	W	D	S	S	chr11
65	CaNIP3;12	CA.PGAv.1.6.scaffold1122.2	NIP	NIP3	W	A	-	K	chr12
66	CaNIP3;13	CA.PGAv.1.6.scaffold1569.5	NIP	NIP3	W	V	A	R	PGAv.1.6.scaffold1569
67	CaSIP1;1	CANN_MIP.chr01.4	SIP	SIP1	-	-	S	S	chr01
68	CaSIP1;2	CA.PGAv.1.6.scaffold245.6	SIP	SIP1	I	-	G	S	chr01
69	CaSIP1;3	CA.PGAv.1.6.scaffold415.6	SIP	SIP1	I	-	P	N	chr03
70	CaSIP1;4	CANN_MIP.chr07.1	SIP	SIP1	-	-	-	-	chr07
71	CaSIP1;5	CA.PGAv.1.6.scaffold960.54	SIP	SIP1	L	-	P	N	PGAv.1.6.scaffold960
72	CaXIP1;1	CA.PGAv.1.6.scaffold564.27	XIP	XIP1	I	T	A	R	chr04
73	CaXIP1;2	CA.PGAv.1.6.scaffold588.13	XIP	XIP1	I	T	A	R	PGAv.1.6.scaffold588

## Data Availability

Not applicable.

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
