# Peer review of "Comparative Analysis of Re-Annotated Genes Provides Insight into Evolutionary Divergence and Expressions of Aquaporin Family in Pepper"

_plants, 2021, doi:10.3390/plants10061039_

Round 1
Reviewer 1 Report
The manuscript presents a descriptive but interesting work about plant aquaporins evolution, with a focus on pepper. This research may be very useful for future functional studies on pepper or other crops. The text is correctly written and results and methods are well-described. Thus, I consider that it can be published almost in present form with few changes.
-L27. The reference used for this sentence is not correct, the manuscript cited is not related to the transport function of aquaporins and briefly mentions this in the introduction. Please add a more appropriate reference.
-Figure 2. Information provided by this figure is very interesting, however, quality image quality is not good enough and phylogenetic tree is too small. Could you please improve this?
-I find the discussion quite brief. In line 230, could you extend this part? Is there any study about this topic in pepper or other crops? Which function could have TIP4 subgroup in this species?
-L243. Could you explain in more detail those functions? in which conditions?
Author Response
Reply to the reviewer comments:
Reviewer #1’s comments:
C1.1. The manuscript presents a descriptive but interesting work about plant aquaporins evolution, with a focus on pepper. This research may be very useful for future functional studies on pepper or other crops. The text is correctly written and results and methods are well-described. Thus, I consider that it can be published almost in present form with few changes. -L27. The reference used for this sentence is not correct, the manuscript cited is not related to the transport function of aquaporins and briefly mentions this in the introduction. Please add a more appropriate reference.
R1.1. We greatly appreciate the reviewer’s comment. We found more appropriate references related to the various transport functions of aquaporins and cited in the Introduction as suggested by the reviewer (line 27).
C1.2. -Figure 2. Information provided by this figure is very interesting, however, quality image quality is not good enough and phylogenetic tree is too small. Could you please improve this?
R1.2. We updated all main figures including the figure 2 in the manuscript with high resolution images and uploaded high quality EPS files when submitting initially, which will replace the figures in the manuscript if our article is accepted. The size of the phylogenetic tree has also been increased as suggested by the reviewer.
C1.3. -I find the discussion quite brief. In line 230, could you extend this part? Is there any study about this topic in pepper or other crops? Which function could have TIP4 subgroup in this species?
R1.3. We further extended the discussion of pepper-specific TIP4 genes in our study (lines 224-230, current lines 242-261). Specifically, we demonstrated pepper-specific evolution of NLR gene family as a similar case for evolution of AQP (TIP4) in pepper (lines 252-261). We also described details for potential function of the pepper genes in TIP4 subgroup through expression and clustering analyses in the Discussion section (lines 277-284)
C1.4.-L243. Could you explain in more detail those functions? in which conditions?
R1.4. We added more details with specific descriptions for known functions of PIP in the Discussion section as suggested by the reviewer (lines 264-269).

Reviewer 2 Report
The authors newly annotated aquaporin family in five different plant species including hot pepper. They also performed a gene expression analysis of AQPs in pepper under abiotic stress conditions. These data are informative but I think the discussion part is insufficient. Please discuss more about novelty of the gene annotation method and physiological significance of TIP4 in pepper. Throughout, resolution of the figure images is low and explanation of the figures is poor, which should be improved.
Major comments:
Table 1: Please write number of newly annotated genes not only in the box of “Total” but also in each box of AQP subfamily.
Figure 1: These graphs are hard to understand for readers. Explanation of Y axes is missing. Font size of numbers is too small. Bars are not well aligned. It is difficult to link the numbers of motifs in the bars to the Table S3. Please explain more carefully in main text and legend.
Figure 2: Fig. 2A, Characters are too small to see. Please magnify double.
Table 2: The alphabets in the column of “Position” seem to represent amino acid residues, but it should be explained on the table.
Figure 5: Please write meaning of time points and color codes in Fig. 5A. Please explain meaning of color code in Fig.5B.
Line218: The authors found newly-annotated AQPs in model plants including rice and Arabidopsis. Please discuss more about these genes and discuss differences from previous studies which did not annotate those genes in rice and Arabidopsis.
Line231: The authors should discuss possible functions/physiological significance of TIP4 subgroup in pepper more in depth.
Minor comments:
Line212: AQPs have various substrates in addition to water. Please rephrase.
Line214: What is annotation of here? Please explain in detail.
Author Response
Reply to the reviewer comments:
Reviewer #2’s comments:
C2.1. The authors newly annotated aquaporin family in five different plant species including hot pepper. They also performed a gene expression analysis of AQPs in pepper under abiotic stress conditions. These data are informative, but I think the discussion part is insufficient. Please discuss more about novelty of the gene annotation method and physiological significance of TIP4 in pepper. Throughout, resolution of the figure images is low, and explanation of the figures is poor, which should be improved.
R2.1. We are grateful for the reviewer’s suggestions for improving our paper. We expanded on the discussion as suggested by the reviewer. The novelty of the target gene family annotation method used in this study was discussed by comparing to the whole gene annotation method in the Discussion part (lines 224-234). Although the actual physiological significance of the TIP4 genes is difficult to discuss without functional studies, expression clustering analysis based on RNA-Seq data could predict the potential function of the TIP4 genes, which we have described in the Discussion part (lines 277-284). We updated all main figures in the manuscript with high resolution images and uploaded high quality EPS files when submitting initially, which will replace the figures in the manuscript if our article is accepted.
Major comments:
C2.2. Table 1: Please write number of newly annotated genes not only in the box of “Total” but also in each box of AQP subfamily.
R2.2. Done (Table 1).
C2.3. Figure 1: These graphs are hard to understand for readers. Explanation of Y axes is missing. Font size of numbers is too small. Bars are not well aligned. It is difficult to link the numbers of motifs in the bars to the Table S3. Please explain more carefully in main text and legend.
R2.3. For better understanding of the readers, we moved Fig. 1B to the supplementary figure1 and updated the figure size, font size, and added explanation of Y-axes for clarity. Explanation of the motif numbers has been added to the legend (lines 110-112, Figure 1 and Supplementary figure 1).
C2.4. Figure 2: Fig. 2A, Characters are too small to see. Please magnify double.
R2.4. Done (Figure 2A).
C2.5. Table 2: The alphabets in the column of “Position” seem to represent amino acid residues, but it should be explained on the table.
R2.5. Done (Table 2).
C2.6. Figure 5: Please write meaning of time points and color codes in Fig. 5A. Please explain meaning of color code in Fig.5B.
R2.6. We added the time units (hours) and explanation of the color codes in Fig. 5 and its legend (lines 206-210 and Figure 5).
C2.7. Line218: The authors found newly-annotated AQPs in model plants including rice and Arabidopsis. Please discuss more about these genes and discuss differences from previous studies which did not annotate those genes in rice and Arabidopsis.
R2.7. Recent studies have reported gene omission in previous annotations even in model plants like Arabidopsis and rice as well as in human genome annotations. Omission of correct gene models in the genome sequences was mainly due to technical limitations of previous whole gene annotation method, and thus advanced methods have emerged to overcome these limitations. Of the latest annotation methodologies, we used the target gene family annotation method to re-annotate AQP genes in the five plant species which allowed us to identify previously omitted genes. We explained the specific differences between previous whole gene annotation method and target gene family annotation method that we used in this study in the Discussion section (lines 224-239)
C2.8. Line231: The authors should discuss possible functions/physiological significance of TIP4 subgroup in pepper more in depth.
R2.8. The potential function of the pepper gene belonging to TIP4 has been added to the Discussion section (lines 277-284). Most of the TIP4 genes have emerged relatively recently and have low expression, which is consistent with a previous report that genes show reduced expression after duplication events (Qian, Wenfeng, et al. 2010). Although their exact functions are difficult to investigate, we discussed the potential function of specific TIP4 gene based on the expression clustering analysis.
Qian, et al. “Maintenance of duplicate genes and their functional redundancy by reduced expression”. Trends Genet (2010).
Minor comments:
C2.9. Line212: AQPs have various substrates in addition to water. Please rephrase.
R2.9. Done (line 216-217).
C2.10. Line214: What is annotation of here? Please explain in detail.
R2.10. We have rewritten the sentence to avoid confusion of readers (lines 224-226).
Reviewer 3 Report
The authors in this manuscript made an effort to provide insight into the evolutionary divergence and expression of the aquaporin family of Pepper. The authors performed the RNA-Seq analysis to show the expression pattern of different AQP genes of the Pepper at different environmental cues.
However, there is a significant lack of information and understanding about the AQP by authors that makes raise serious doubt about the study.
Here are few major concerns:
- The authors tried to re-annotate the AQP genes of Arabidopsis and rice without producing any convincing reasons. The AQP family in plants is well annotated based on several phylogenetic studies and the composition of ar/R region amino acids.
- The authors also mentioned 14 motifs of the AQP without providing any background about it. To this reviewer’s understanding, there is no such motif classification for plant AQPs. If authors think otherwise, then they need to provide information about it.
- The phylogenetic tree provided in the manuscript is poorly constructed and lacks the information that a reader needs to understand the manuscript better.
- Discussion is poorly written, and both introduction and discussion lack citation of some seminal works in this field.
- The authors lack basic knowledge of plant aquaporins that triggers the panic in this reviewer.
For example, the NIP family of the plants are grouped into four subfamilies which are well known in the plant aquaporin community (check the paper Abascal et al., 2014). Again, Arabidopsis has a total of 35 AQPs and nine NIP family members.
- The subclassification of NIPs of Arabidopsis is wrong. According to the community, the NIP I subfamily has tryptophan at the H2 position of the ar/R region, while NIP II has Alanine.
- The NIP 7;1 has Alanine at H2 position, not Cyst as mentioned in the suppl table S2.
- The amino acids of NIP1;1 and NIP1;2 of Arabidopsis at LE1 and LE2 position of the Ar/R region are missing. This reflects the lack of knowledge of authors on aquaporins. It also raises doubt about multiple sequence alignment performed for the study and hence its phylogenetic tree.
Author Response
Reply to the reviewer comments:
Reviewer #3’s comments:
C3.1. The authors in this manuscript made an effort to provide insight into the evolutionary divergence and expression of the aquaporin family of Pepper. The authors performed the RNA-Seq analysis to show the expression pattern of different AQP genes of the Pepper at different environmental cues. However, there is a significant lack of information and understanding about the AQP by authors that makes raise serious doubt about the study. Here are few major concerns: The authors tried to re-annotate the AQP genes of Arabidopsis and rice without producing any convincing reasons. The AQP family in plants is well annotated based on several phylogenetic studies and the composition of ar/R region amino acids.
R3.1. We appreciate the reviewer’s comments. AQP gene family has been studied and well-annotated in various crops, including Arabidopsis and rice. However, re-annotation is necessary even for model plants as suggested by recent reports of missing genes occurring in whole gene annotation, which can bias further downstream functional and comparative analysis. Therefore, in this study, we performed re-annotation using an updated annotation tool based on target gene methodology which first selects regions with a target domain within an assembled genome, and then proceeds with intensive multi-step annotation processes for those regions. This allows identification of previously omitted AQP genes in five species, including model species like Arabidopsis and rice as well as in pepper. Further details are written in the Discussion section (lines 224-239).
C3.2. The authors also mentioned 14 motifs of the AQP without providing any background about it. To this reviewer’s understanding, there is no such motif classification for plant AQPs. If authors think otherwise, then they need to provide information about it.
R3.2. Conserved sequences such as the AQP domain play important biological roles in physiological processes and provide fundamental information to the evolution of the gene family. Thus, motif analysis was performed to detect conserved or variable sequences of the AQP domains. The motif structures of AQP domains provide structural characteristics of the AQP domain.
C3.3. The phylogenetic tree provided in the manuscript is poorly constructed and lacks the information that a reader needs to understand the manuscript better.
R3.3. In this study, we constructed the phylogenetic tree using IQ-Tree version 2 reported in Minh et al., 2020 (Molecular Biology and Evolution, cited 534 times), which is a very popular, latest advanced method for performing phylogenetic studies. If the reviewer has further specific suggestions to improve our tree, we can improve accordingly.
Minh et al. "IQ-TREE 2: New models and efficient methods for phylogenetic inference in the genomic era." Molecular biology and evolution 37.5 (2020)
C3.4. Discussion is poorly written, and both introduction and discussion lack citation of some seminal works in this field.
R3.4. We have improved our manuscript, especially the Discussion section, mainly considering specific comments from reviewer 1 and 2. If the reviewer provides specific comments for further improvement, we will revise accordingly.
C3.5. The authors lack basic knowledge of plant aquaporins that triggers the panic in this reviewer. For example, the NIP family of the plants are grouped into four subfamilies which are well known in the plant aquaporin community (check the paper Abascal et al., 2014). Again, Arabidopsis has a total of 35 AQPs and nine NIP family members.
R3.5. The number of AQP genes from NIP subfamily in the previous study of Arabidopsis is nine, which we have specifically written in Table S2, and two additional NIP genes present in the genome were detected through re-annotation from our study. As a result, a total of 41 (38 public genes, 3 newly identified genes) was predicted by functional annotation based on Pfam (version 30.0) analyses. Generally, gene families are classified via functional annotation based on domain analysis. However, functional annotation methods are also advanced, and the number of multi-copy gene families is variable because they are sensitive to methods, tools, and parameters used. For example, the representative multi-copy gene family, NLR, is also reported with a slightly different number of genes due to different methods; Andolfo et al., reported the number of NLR in tomato genome was 294, whereas Seo et al., revealed that the number of same gene family is 267 in the same genome. In this study, we conducted classification of AQP gene family based on the most advanced method and then followed the subfamily classification rules of the prior studies as cited in our manuscript.
Andolfo, et al. "Overview of tomato (Solanum lycopersicum) candidate pathogen recognition genes reveals important Solanum R locus dynamics." New Phytologist 197.1 (2013).
Seo, et al. "Genome-wide comparative analyses reveal the dynamic evolution of nucleotide-binding leucine-rich repeat gene family among Solanaceae plants." Frontiers in plant science 7 (2016).
C3.6. The subclassification of NIPs of Arabidopsis is wrong. According to the community, the NIP I subfamily has tryptophan at the H2 position of the ar/R region, while NIP II has Alanine.
R3.6. When we compared the subfamily to which Arabidopsis NIP genes belonged with subfamilies reported by Abascal et al., 2014, the genes assigned to specific subgroups were identical. The genes assigned to the NIPI subfamily in Abascal's paper were assigned to the NIP2 subgroup in our study, and the genes assigned to the NIP II were assigned to the NIP1 subgroup. In this study, the genes were divided into 11 subgroups in total, and if the NIP was to be divided into four subgroups, we should separate other families considering more specific conditions. Finally, the number of subgroups will inevitably increase. there were papers that reported two or three subgroups of NIP subfamilies same with our study (Wallace and Daniel 2004; Deshmukh et al., 2015).
Wallace and Daniel. "Homology modeling of representative subfamilies of Arabidopsis major intrinsic proteins. Classification based on the aromatic/arginine selectivity filter." Plant Physiology 135.2 (2004).
Deshmukh, et al. "A precise spacing between the NPA domains of aquaporins is essential for silicon permeability in plants." The Plant Journal 83.3 (2015).
C3.8. The NIP 7;1 has Alanine at H2 position, not Cyst as mentioned in the suppl table S2.
R3.8. As NIP7;1, we used alternative splicing forms of the same loci, which generated the differences as mentioned by the reviewer. Thus, we replaced AT3G06100.3 to AT3G06100.1 having Alanine without Cysteine. (Table S2)
C3.9. The amino acids of NIP1;1 and NIP1;2 of Arabidopsis at LE1 and LE2 position of the Ar/R region are missing. This reflects the lack of knowledge of authors on aquaporins. It also raises doubt about multiple sequence alignment performed for the study and hence its phylogenetic tree.
R3.9. As mentioned above in R3.8, the NIP1;1 and NIP1;2 genes mentioned by the reviewer are different alternatively spliced forms of the same loci which caused the problem. Thus, we replaced AT4G18910.2 to AT4G18910.1 and AT4G19030.3 to AT4G19030.1 having Alanine and Arginine mentioned by the reviewer. (Table S2)
Round 2
Reviewer 3 Report
- As the authors mentioned in their reply, it is essential to reannotate the genome with new techniques; however, in the particular study, newly identified genes through reannotation did not come up with better results, particularly in the case of Arabidopsis. To annotate a gene as aquaporins, the gene should be translated to a full-length protein of six transmembrane helices with five loops. A quick examination of some of the newly annotated Arabidopsis genes by this study showed that these genes are translated into truncated proteins. Like AT1G52180 (~124 amino acids) and AT2G16835 (65 amino acids). Therefore, these genes are annotated as aquaporin-like proteins by TAIR. These genes may not be translated into protein and could act as regulatory transcripts. This clearly shows the poor annotation work of the study. Also, the manuscript lacks the details of annotation criteria and better explanation. For the study, the reannotation of the genomes other than pepper is not essential, to this reviewers understanding. The annotation of pepper genes and its expression study are perhaps enough for the publication.
- About the domain, it is pretty confusing. The authors did not mention what are the domains they are talking about in the manuscript. It would have been better if authors have shown the region of the protein sequence in a multiple sequence alignment that they consider as a domain. Where are these 14 domains located in the secondary structure of the aquaporins? What are the criteria the author used to select these 14 domains? The manuscript lacks this information. If authors are referring these 14 domains to any previous studies, they should cite the study in the context. Again, the authors suggesting these domains are essential for the transport activity of the aquaporins; how they come to this conclusion? Have they tested each domain to see their role in transport activity, or are they referring to any previous study or speculating?
- Phylogenetic tree analysis: the authors sub-classified NIPs into two groups. However, based on the phylogenetic tree, the NIPs are sub-divided into at least three subgroups. The authors’ subgrouping numbering of aquaporins subfamilies in phylogenetic does not match the subgroup number in table 2. These subgroup numbering should be consistent.
- In their reply no R3.8, the authors mentioned they used AT3g06100.3 spliced variants in their previous annotation. However, they should know that variant3 of NIP7;1 is transcript variants that lack full-length protein. Same issue with reply R3.9. This again raises doubt on annotation performed in this study.
Other issues of the manuscript:
- In table 2, pepper aquaporins PIP1, PIP2, and PIP3 subfamily members have the same ar/R amino acid compositions but are renamed differently. As per the text, they should be named PIP in the subfamily column rather than PIP1,2 and 3.
- The authors failed to explain how they sub-grouped these aquaporins mentioned in Table 2 and S2 from 1-11. The reviewer assumes this numbering is based on the phylogenetic tree. But this is not explained anywhere in the text or table.
- In table 1, what is the number in the parenthesis? The authors need to explain the table in the table legend clearly and also in the text too.
- In line no 83, the authors mentioned the amino acid compositions of the ar/R filter and cite Tables 1-2, but in table 1 no such amino acids are mentioned—this needs to be corrected.
- In Suppl Table 2, what the asterisk in gene name suggests? The figure legend of Table 2 should be expanded to explain the table adequately without leaving it at the mercy of the reader to speculate.
- In Table S3, the authors showed around 50 motifs, what those motifs are, and their significance. The Suppl table S3 needs a better table description. What is length, position, and the number of motifs columns signifies? These need explanation in table legends and also in the result section.
- Again, what are the positions in figure 1 and suppl figS1? Are they similar to the motifs? Need explanation. Are the positions and domains are interchangeable?
- In Figure 2, the authors suggested the gene name mentioned in the phylogenetic tree are known functional genes. However, is that mean only these genes are functionally characterized? For example, most Arabidopsis aquaporins (NIPs) are functionally characterized but not mentioned in this figure.
- In figure 5, (A) what is the basis of clustering the DEG for each stress condition, and what is membership grade? These need better explanation in the results and, if possible, in the figure legends. In figure 5B, what is the number in the Y-axis of the heatmap, no of genes? Figure 5C, there is more than ten GO mentioned in the X-axis. What is % of Hit? Please explain in the figure legend.
- In suppl. fig 1, the figure heading does not match the figure and its figure legend. Also, the Y-axis labeling of XIP moved to the NIP.
- The figure legends of suppl figure 2 need more description so that reader does not have to depend upon the main text heavily. Such as how many subfamilies of MIPS are found in which genome.
- More importantly, the authors should reconsider the color coordination of figures keeping color-blindness in mind.
Author Response
Please check the attached file for fully answered revision comments.

Round 3
Reviewer 3 Report
The authors have satisfactorily addressed all the issues in the manuscript.